# Observation of an Alice ring in a Bose–Einstein condensate

**Alina Blinova** [1,2] ✉, **Roberto Zamora-Zamora** [3,5], **Tuomas Ollikainen** [1,3,6], **Markus Kivioja** [4], **Mikko Möttönen** [3] & **David S. Hall** [1]

Monopoles and vortices are fundamental topological excitations that appear in physical systems spanning enormous scales of size and energy, from the vastness of the early universe to tiny laboratory droplets of nematic liquid crystals and ultracold gases. Although the topologies of vortices and monopoles are distinct from one another, under certain circumstances a monopole can spontaneously and continuously deform into a vortex ring with the curious property that monopoles passing through it are converted into anti-monopoles. However, the observation of such Alice rings has remained a major challenge, due to the scarcity of experimentally accessible monopoles in continuous fields. Here, we present experimental evidence of an Alice ring resulting from the decay of a topological monopole defect in a dilute gaseous $^{87}$Rb Bose–Einstein condensate. Our results, in agreement with detailed first-principles simulations, provide an unprecedented opportunity to explore the unique features of a composite excitation that combines the topological features of both a monopole and a vortex ring.

Symmetry-breaking phase transitions are ubiquitous in physics[1], appearing in contexts as diverse as the cooling of the early universe, the emergence of ferromagnetism, and the onset of super-conductivity. As a phase transition proceeds, uncorrelated domains of the new phase grow and assume preferred field configurations where they meet. Topological defects appear where no uniform configuration can unite the domains, adopting the form of surfaces (walls), lines (strings and vortices), and singular points (monopoles). Strings and monopoles carry conserved topological charges which, depending on the physical properties of the system, can manifest as magnetic, electric, or even quark colour charges[2].

The Alice string[3,4] is unusual among topological defects, appearing in certain grand unified theories[2,5,6] as an element that converts a monopole into an anti-monopole as it travels around the string[7]. This fascinating "looking-glass" property, which gives the excitation its name[8], has a direct counterpart in condensed matter systems[9–11],

where Alice strings have been identified with half-quantum vortices in superfluids[12,13] and $\pi$-disclinations in nematic liquid crystals[11,14].

Alice strings are intimately tied to monopoles in a second surprising way. At the monopole singularity itself, the field has no single well-defined configuration and is therefore required to vanish. The energy cost associated with the recovery of the field determines the characteristic size of the region over which the system heals to the phase that supports the defect. However, within such depleted singular regions another phase with a different symmetry may be present, and consequently the system energy can be reduced by adopting more exotic topological configurations. A monopole can thus become energetically unstable against a deformation into a closed loop of Alice string, i.e., an Alice ring, which preserves the topology of the monopole field far from its core. Such monopole core deformations have been predicted in nematic liquid crystals[15–18], in 't Hooft–Polyakov monopoles[19,20] within the field theory of Alice

---

[1]Department of Physics and Astronomy, Amherst College, Amherst, MA 01002–5000, USA. [2]Department of Physics, University of Massachusetts Amherst, Amherst, MA 01003, USA. [3]QCD Labs, QTF Centre of Excellence and InstituteQ, Department of Applied Physics, Aalto University, P.O. Box 13500, FI–00076 Espoo, Finland. [4]Faculty of Information Technology, University of Jyväskylä, P.O. Box 35, FI-40014 Jyväskylä, Finland. [5]Present address: Quanscient Oy, Tampere, Finland. [6]Present address: Institut für Experimentalphysik, Universität Innsbruck, Technikerstraße 25, 6020 Innsbruck, Austria. ✉e-mail: aliblinova@gmail.com

**Fig. 1 | Schematic illustration of a topological monopole defect and its decay product, an Alice ring, in a polar-phase Bose–Einstein condensate.**
**a** Topological monopole with a zero-density core (black dot) and the corresponding polar order parameter field represented by the director $\hat{\mathbf{d}}$ (white cones) and the scalar phase $\varphi$ (background colour). Such a monopole can be created by adiabatically bringing the zero point of the quadrupole magnetic field (blue connected arrows) into the centre of the spheroidal condensate. For clarity, the condensate is shown in section for $y > 0$. **b, c** Order parameter field after the monopole has dynamically decayed into an Alice ring filled with the ferromagnetic phase (red ring). The polar condensate is shown in section for $y > 0$ (**b**) and $z < 0$ (**c**).

electrodynamics[21,22], and more recently in spinor Bose–Einstein condensates (BECs)[23,24].

In this work, we present experimental evidence of an Alice ring in a spin-1 Bose–Einstein condensate. The Alice ring appears during the time evolution of a topological monopole defect[25] in the polar magnetic phase of the BEC, where it takes the form of a vortex ring filled with superfluid in the ferromagnetic phase[23] (Fig. 1). Experimental images, in good agreement with numerical simulations, reveal that after 5 ms of evolution the initial monopole decays into an extended spin structure consistent with the analytical expectation for an Alice ring (Figs. 2 and 3). Interestingly, both the experiment and the numerical simulations reveal that an initially off-centred monopole defect evolves into a tilted Alice ring (Fig. 4), dramatically underscoring its presence by enhancing the visibility of its ferromagnetic core.

## Results

Within the mean-field approximation[26], a spinor condensate order parameter may be expressed in terms of the atomic density $n$ and the normalised spinor $\zeta$ as

$$\Psi(\mathbf{r}, t) = \sqrt{n(\mathbf{r}, t)}\zeta(\mathbf{r}, t) \qquad (1)$$

For a spin-1 condensate, $\zeta \equiv (\zeta_{+1}, \zeta_0, \zeta_{-1})^{\mathrm{T}}$ in the Zeeman basis with complex-valued components $\zeta_m$ indexed by the magnetic quantum number $m$. We ignore effects beyond the standard mean-field approximation, such as those related to finite temperature.

The BEC can exhibit magnetic ordering that breaks the full symmetry of the Hamiltonian[27] (Methods), leading to states with distinct symmetries and corresponding magnetic phases. Depending on the type of spin–spin interactions, the ground-state magnetic phase of a spin-1 BEC may be ferromagnetic (FM), in which the local average spin $|\langle \mathbf{F} \rangle|$ is maximised, or polar, in which the local spin vanishes and the order is nematic. More generally, the condensate can exist in a mixed phase that possesses both FM and polar order (Methods), of which the pure phases are the limiting cases.

Monopoles and Alice rings are supported in the polar phase, where the polar spinor can be expressed in the Cartesian basis[26] as $\zeta_\mathrm{P}(\mathbf{r}, t) = e^{i\varphi(\mathbf{r}, t)}\hat{\mathbf{d}}(\mathbf{r}, t)$ in terms of the scalar phase $\varphi$ and the director $\hat{\mathbf{d}} = (d_x, d_y, d_z)$. The director is a three-dimensional real-valued unit vector that conveniently specifies the local quantization axis along which only the $m = 0$ spinor component is populated. Importantly, the polar order parameter is nematic (Methods) since it is invariant under the simultaneous substitutions $\hat{\mathbf{d}} \rightarrow -\hat{\mathbf{d}}$ and $\varphi \rightarrow \varphi + \pi$. Thus we need to define the scalar phase only on the interval $[0, \pi)$, in which case the director assumes all values on the unit sphere.

The director field of a monopole is illustrated in Fig. 1a. Since the energy cost of forcing the particle density to zero at the singular point is relatively high, it is energetically favourable for the point defect to deform into a ring that fills with superfluid in the ferromagnetic phase, as first pointed out by Ruostekoski and Anglin[23] (Fig. 1b, c). Along any poloidal curve, such as $\mathcal{L}$ in Fig. 1b, the vector $\hat{\mathbf{d}}$ undergoes a $\pi$ rotation while the scalar phase $\varphi$ changes continuously by $\pi$. At the location where oppositely oriented $\hat{\mathbf{d}}$ vectors meet, the continuity of the order parameter is preserved by a $\pi$ phase jump. Although the location of the phase jump and director reversal is a matter of gauge choice, a monopole traveling a path that encircles the ring poloidally must encounter it at some point and turn into an anti-monopole, consistent with the action of an Alice ring[9,28]. Moreover, the additional continuous $\pi$ phase winding along $\mathcal{L}$ shows that the Alice ring manifests as a half-quantum vortex (HQV) ring in the BEC[23].

Our experiment begins with the creation of a topological monopole defect in an optically trapped BEC[25]. Initially, the $\hat{\mathbf{d}}$ field is aligned with a uniform bias magnetic field $\mathbf{B}_\mathrm{b}(t)$ pointing in the $+z$ direction. An additional quadrupole field contribution $\mathbf{B}_\mathrm{q}(\mathbf{r}, t) = (x\hat{\mathbf{x}} + y\hat{\mathbf{y}} - 2z\hat{\mathbf{z}})b_\mathrm{q}(t)$ of strength $b_\mathrm{q}$ is introduced such that the total field is

$$\mathbf{B}(\mathbf{r}, t) = \mathbf{B}_\mathrm{b}(t) + \mathbf{B}_\mathrm{q}(\mathbf{r}, t) \qquad (2)$$

where the origin of the coordinate system coincides with the centre of the BEC (Methods). The field zero, defined as the point at which the magnetic field vanishes, rests above the condensate along the $z$ axis, and all directors remain well-approximated by $\hat{\mathbf{d}} = \hat{\mathbf{z}}$. The monopole is introduced by slowly reducing the bias field strength $B_\mathrm{b} \rightarrow 0$, bringing the field zero into the centre of the condensate. The directors, precessing at the local Larmor frequency, follow the local field direction nearly adiabatically as the bias field is reduced. The result is the monopole field configuration $\hat{\mathbf{d}}_\mathrm{m} = (x, y, -\bar{z})/\bar{r}$, written here in terms of the rescaled $z$ coordinate $\bar{z} = 2z$ and $\bar{r} = \sqrt{x^2 + y^2 + \bar{z}^2}$.

Immediately following the creation process, $\mathbf{B}_\mathrm{q}$ is extinguished while $\mathbf{B}_\mathrm{b}$ is rapidly increased to 1.2 G along a direction of our choice, $\hat{\mathbf{p}}$, which defines the quantization axis for the following evolution and imaging. The field during the evolution is strong enough to ensure that the ground-state magnetic phase is polar (Methods). The condensate subsequently evolves in the uniform magnetic field for a time $T$, at the end of which the condensate is released from the optical trap. We apply a brief magnetic-field gradient during the subsequent free fall and expansion of the condensate, which spatially separates the three spinor components according to their magnetic quantum number $m$ along $\hat{\mathbf{p}}$. The column densities of the expanded spinor components are then imaged absorptively along the $y$ (side) and $z$ (top) axes.

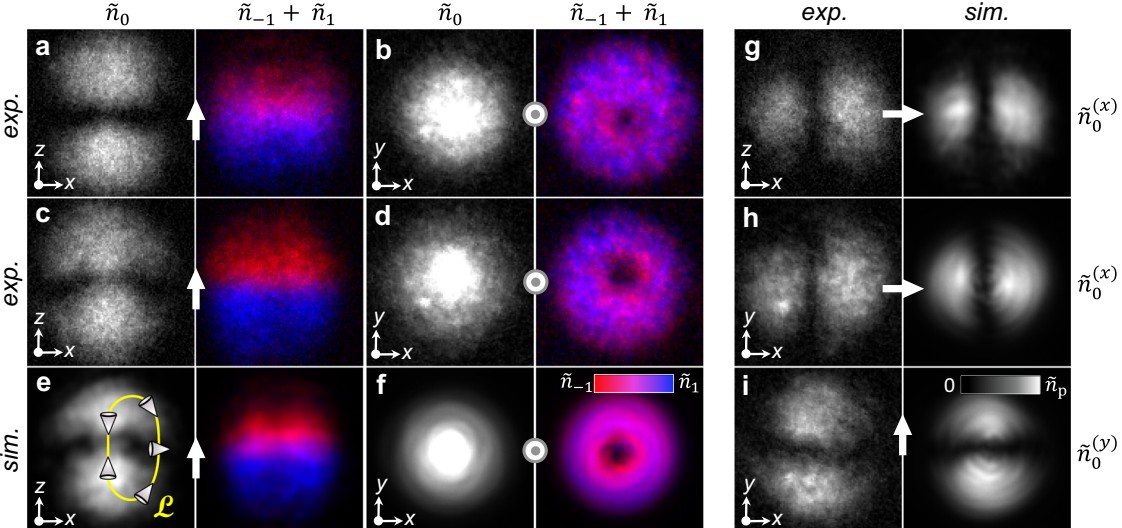

**Fig. 2 | Topological monopole and Alice ring in a spin-1 Bose–Einstein condensate.** In all panels, we show particle column densities $\tilde{n}_m$ for the $m = 0$ (white), $m = +1$ (blue), and $m = -1$ (red) spinor components projected along $\hat{\mathbf{p}}$ (white arrows between corresponding subpanels). **a, b** Experimentally-obtained images of the monopole projected along $\hat{\mathbf{p}} = \hat{\mathbf{z}}$ at $T = 0$ ms and viewed from the side (along $y$) (**a**) and from the top (along $z$) (**b**). **c, d** As panels (**a**) and (**b**) but for $T = 5$ ms, showing in **c** the emergence of a well-defined column of $m = 0$ atoms along the $z$ axis. **e, f** Results of three-dimensional first-principles numerical simulations

corresponding to (**c**) and (**d**), with schematic cones representing $\hat{\mathbf{d}}$ along a continuous loop $\mathcal{L}$. Where $\hat{\mathbf{d}}$ is parallel to $\hat{\mathbf{p}}$, the atoms are entirely in the $m = 0$ component, whereas $\hat{\mathbf{d}}$ perpendicular to $\hat{\mathbf{p}}$ results in atoms solely in the $m = \pm 1$ components. **g–i** Experimental (left) and simulated (right) particle column densities at $T = 5$ ms for $\hat{\mathbf{p}} = \hat{\mathbf{x}}$ (**g, h**) and $\hat{\mathbf{p}} = \hat{\mathbf{y}}$ (**i**), demonstrating an absence of the $m = 0$ column along these directions. For each panel, the field of view is $219 \times 219\ \mu m^2$ and the peak column density is $\tilde{n}_p = 6.89 \times 10^8$ cm$^{-2}$. Densities $\tilde{n}_0 > \tilde{n}_p$ appear white.

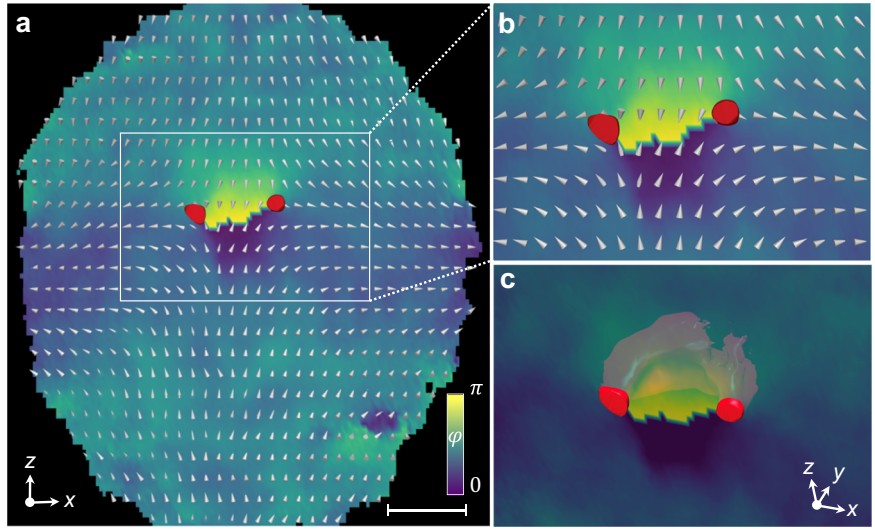

**Fig. 3 | Detailed simulation of the Alice ring emerging from a monopole.** **a, b** Cross-section ($y = 0$ plane) of a released and expanded BEC for $T = 4$ ms, with $\hat{\mathbf{d}}$ and $\varphi$ of the polar phase represented by white cones and a background colour map, respectively. The FM phase is shown in red for $|\langle \mathbf{F} \rangle| \geq 0.95$. The region bounded by the white box is approximately $95\ \mu m$ by $60\ \mu m$ and shown magnified in (**b**). The scale bar denotes $30\ \mu m$. **c** As (**a, b**) but rotated into an isometric view with a semi-

transparent phase colour map in order to reveal the structure of the FM ring, shown in section for $y > 0$. The diameter of the expanded FM ring is $30\ \mu m$. Some fluid in the FM phase appears throughout the condensate at densities smaller than that of the nearly pure FM ring, and we extract the polar director and phase using the technique described in the Methods.

The experimentally measured spinor component density profiles of the monopole created using this technique for $T = 0$ and $\hat{\mathbf{p}} = \hat{\mathbf{z}}$ are shown in Fig. 2a, b. The monopole is located at the centre of the condensate, where the particle density nominally vanishes. In Fig. 2a, the side view reveals that the $m = 0$ spinor component has a solitonic structure with disconnected top ($z > 0$, $d_z < 0$) and bottom ($z < 0$, $d_z > 0$) lobes. The overlapping $m = \pm 1$ components fill the region between the lobes in the vicinity of the $xy$ plane ($z = 0$, $d_z = 0$) and

contain density holes along the $z$ axis which correspond to singly quantized vortices of opposite circulation (Fig. 2b). These are the expected component density profiles of a monopole with the quadrupolar director field of Fig. 1a. Similar density profiles, related by rotations, have been previously observed for $\hat{\mathbf{p}}$ chosen along different directions, all sharing the lobe and vortex structures in the $m = 0$ and $m = \pm 1$ spinor components with respect to a chosen $\hat{\mathbf{p}}$-dependent basis[25]. In particular, at every point the $m = 0$ spinor component

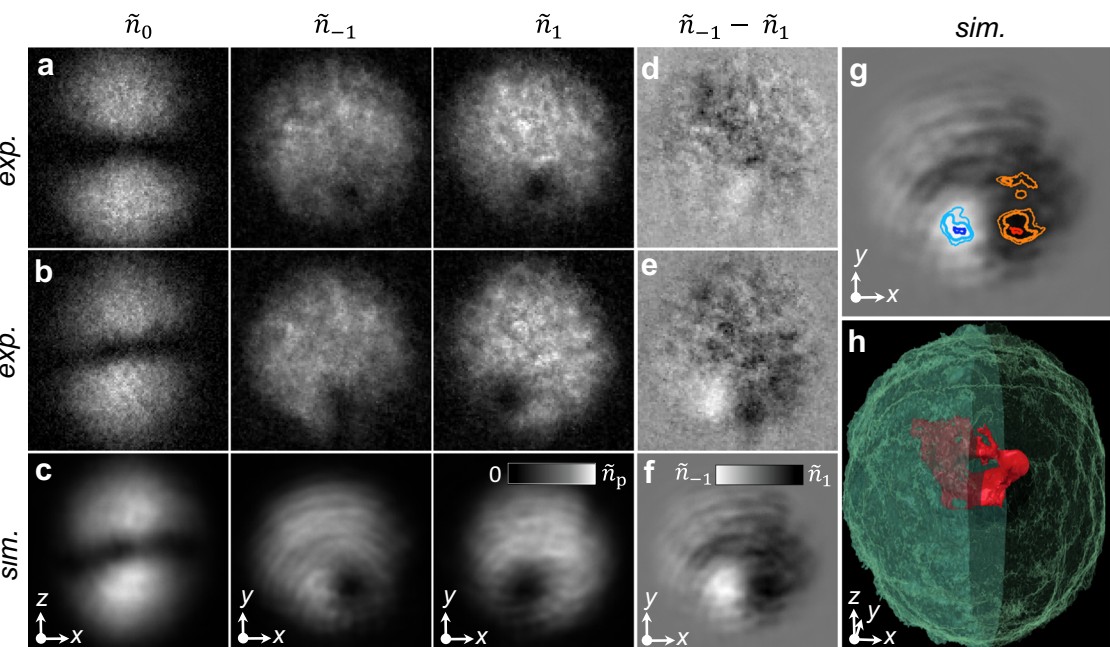

**Fig. 4 | Evidence of the Alice ring near the edge of the condensate.**
**a**–**c** Experimental (**a**, **b**) and simulated (**c**) images of the $m = -1, 0, 1$ component column densities $\tilde{n}_m$, projected along $\hat{z}$, for a monopole displaced in the $-y$ direction, as viewed from the top and side for $T = 0$ ms (**a**) and $T = 4$ ms (**b**, **c**). **d**–**f** Experimental (**d**, **e**) and simulated (**f**) difference in the $m = \pm 1$ column densities, $\tilde{n}_{-1} - \tilde{n}_1$, with $\hat{p} = \hat{z}$ for $T = 0$ ms (**d**) and $T = 4$ ms (**e**, **f**). The monopole decay product exhibits longitudinal magnetisation within the disk-shaped region formed by the $m = \pm 1$ components. Initially overlapping density holes in the $m = \pm 1$ spinor components in (**d**) become more separated with time in (**e**, **f**). **g** As panel (**f**), but with additional contours such that light blue and orange denote the greatest column densities of $m = -1$ and $m = +1$ atoms, respectively, and dark blue and red indicate ferromagnetic regions where $|\langle \mathbf{F} \rangle| \geq 0.95$ with $-\hat{z}$ and $+\hat{z}$ magnetisation, respectively. **h** Simulated polar condensate (green colour), shown in section for $x < 0$, and the emergent ferromagnetic region $|\langle \mathbf{F} \rangle| \geq 0.95$ (red) in the shape of a ring that is bent upwards near the condensate boundary. Small disconnected ferromagnetic regions are excluded for clarity. For each panel, the field of view is $219 \times 219 \ \mu m^2$ and the peak column density is $\tilde{n}_p = 6.89 \times 10^8 \ cm^{-2}$.

density is proportional to the square of the component of the director lying along $\hat{p}$.

We repeat the experiment for different evolution times $T$ after the monopole creation to map the dynamics of the monopole configuration (Fig. 2). In Fig. 2c–f, we compare the component density images in all three projections $\hat{p} = \hat{x}, \hat{y}, \hat{z}$ with component densities obtained through first-principles simulations (Methods) and find good agreement. This agreement not only validates our experimental results but also provides us with the opportunity to study numerically the detailed decay dynamics and the properties of the order parameter that are not directly observable experimentally.

By $T = 5$ ms a column of $m = 0$ (polar) atoms appears in the experimental images, connecting the top and bottom $m = 0$ lobes along the $z$ axis (Fig. 2c). No comparable connecting columns are observed for experiments with $\hat{p} = \hat{x}$ and $\hat{p} = \hat{y}$ (Fig. 2g–i), which show instead separated $m = 0$ lobes, consistent with the spinor component densities of the initial monopole[25]. Note that the column of $m = 0$ atoms in the $\hat{p} = \hat{z}$ projection (visible along the $z$ axis in Fig. 2c) is invisible in the auxiliary projections as these same atoms get projected into an equal superposition of $m = \pm 1$. The appearance of a polar column along the $z$ axis is also independent of the direction from which the zero of the magnetic field is brought to the centre of the condensate (Supplementary Fig. 1), reflecting instead the rotational symmetry of the trapping potential about the $z$ axis.

As the $m = 0$ polar column appears for $\hat{p} = \hat{z}$, the $m = \pm 1$ spinor components retain their initial vortical density profiles and appear essentially unchanged as viewed along the $z$ axis (Fig. 2d, f). However, they drift apart axially, leading to two partially polarised mixed-phase regions of opposite longitudinal magnetisation that smoothly join the pure polar phase in the vicinity of the $xy$ plane (Fig. 2c, e). Our simulations confirm that these partially polarised regions result from differential magnetic forces on the $m = \pm 1$ spinor components during the quench of the magnetic gradient field coils (Supplementary Fig. 2). Although the topological stability of the monopole excitation cannot be strictly guaranteed here, the director and scalar phase associated with the polar order remain well-defined (Methods), and the appearance of the mixed-phase regions does not in itself pose an existential threat to the underlying topology of the polar phase. The principal new feature marking the topological changes to the polar order within the condensate is therefore the column of polar ($m = 0$) atoms along the $z$ axis.

Since the atoms in the connecting column are in the pure polar phase, a point singularity no longer resides at the centre of the BEC. Nevertheless, a singularity in the polar phase must exist somewhere within the condensate, assuming the director field of the initial monopole is preserved at the condensate boundary. Note that simple initial displacements of the monopole singularity away from the centre result in a significant change in the symmetry of the density profiles of each spinor component (Supplementary Fig. 3): a horizontal displacement results in a horizontal shift of the $m = \pm 1$ vortex centres, and a vertical displacement results in a vertical shift of the solitonic gap between the $m = 0$ lobes. We observe neither of these shifts in the density profiles of Fig. 2a, c, leading us to conclude that the polar ($m = 0$) column is not explained by a simple displacement of the initial point singularity.

To identify the nature of the singularity and resolve this puzzle, we consider the directors along a closed curve $\mathcal{L}$, as shown in Fig. 2e. In traversing the curve from the top of the condensate to the bottom along its boundary, the directors rotate by $\pi$ due to the monopole imprinting process[25]. As a result, the directors along the $z$ axis in the top $m = 0$ lobe are antialigned with their counterparts in the bottom $m = 0$ lobe. In completing $\mathcal{L}$ along the column, continuity of the order

parameter therefore requires there to be a $\pi$ discontinuity in the scalar phase where the oppositely oriented directors meet, as illustrated in Fig. 1b. This phase discontinuity is compensated by a continuous $\pi$ phase winding along $\mathcal{L}$. The presence of the column and the retention of the monopole boundary conditions thus suggest that the initial topological point singularity has deformed to a ring that circumscribes the $m = 0$ column, with a half quantum of poloidal circulation arising from the spatial variation in the scalar phase[26]. Our experimental observations thus match the description of the Alice ring[23].

Our simulation results, shown in Fig. 3, directly confirm the anticipated poloidal $\pi$ rotation and winding of $\hat{\mathbf{d}}$ and $\varphi$, respectively. These features are centred on a prominent ferromagnetic torus (with $|\langle\mathbf{F}\rangle| > 0.95$) that provides an alternative means of identifying the location of the emergent Alice ring. The FM ring results from the conversion of the polar phase to the ferromagnetic phase within the core of the Alice ring, since this is energetically less costly than forcing the particle density to vanish at this singular defect of the polar order. The extent of the FM ring is determined by the spin healing length[23], $\xi_s \approx 2.5\,\mu m$, roughly one third of the size of the condensate and a conveniently observable size for imaging. Interestingly, the direction of the scalar-phase winding about the Alice ring is determined by the direction from which the field zero enters the condensate and has direct consequences for what we observe in the spinor component densities (Supplementary Fig. 1).

Next, we demonstrate that a displacement of the initial location of the monopole in the experimental procedure yields an extraordinary view of the cross-section of the Alice ring as it develops, revealing its half-quantum of circulation and the ferromagnetic core. The displacement results from an intentional addition of a small $y$ component to the bias field $\mathbf{B}_b$ during the monopole creation ramp. The subsequent monopole evolution is quite different from the centred case: almost immediately the vortex cores in the $m = \pm 1$ spinor components move away from one another, each filling with the other component as shown in Fig. 4b, c (see also Supplementary Fig. 4). Since the $m = 0$ spinor component essentially vanishes in the $xy$ plane where the $m = \pm 1$ spinor components are most pronounced, the filled-core vortices can be interpreted in the plane as a half-quantum vortex dipole (Methods) with filled ferromagnetic cores of opposite magnetisation (Fig. 4d, e). A planar cross section of a vortex ring presents as a vortex dipole; hence our experimental observations are consistent with an Alice ring bending up through the $xy$ plane, likely due to its interaction with the condensate boundary[29]. The bending of the Alice ring and the development of the ferromagnetic domains are confirmed by our numerical simulations, as illustrated in Fig. 4f–h. The simulations also reveal the presence of significant toroidal magnetisation within the ring core[23].

The characteristic features of the Alice ring, as discussed above, remain identifiable in experimental images of the condensate for evolution times of up to ≈10 ms. A close analysis of the director field and scalar phase in the numerical simulations suggests, however, that the Alice ring survives under our experimental conditions up to a time scale of 100 ms (Supplementary Fig. 5).

## Discussion
Our experimental evidence and numerical analysis lead to the long-awaited conclusion that Alice rings exist in nature. Concurrently, we report an experimental technique to verifiably create Alice rings in an ultracold quantum gas. This unprecedented level of topological engineering together with our initial indications of unexpectedly long-lived Alice rings may enable the future demonstration of the charge conjugation of monopoles that pass through Alice rings. Such an experiment calls for multiple monopole defects including both positive and negative topological charges, a scenario recently studied in the case of Dirac monopoles[30]. Furthermore, our techniques can be directly applied to investigate the decay of monopoles in BECs of $^{23}$Na, where

the existence of two stable Alice ring solutions has recently been predicted[31].

Analogues of the Alice ring may also appear in systems with additional magnetic phases. For example, the $F = 2$ spinor BEC has two nematic phases, both of which support monopole solutions. In the absence of a magnetic field these two phases are energetically degenerate at the mean-field level. Fortunately, the introduction of quantum fluctuations[32–34] or a quadratic Zeeman shift arising from a magnetic field[24,26] lifts this continuous degeneracy, enhancing their distinguishability and individual addressability. Monopoles in the uniaxial nematic phase are topologically stable and similar to those in the polar phase, but any emergent Alice ring would have an integer (rather than half-integer) phase winding. Monopoles in the biaxial nematic phase are not topologically stable, yet they can still support Alice ring solutions[24]. Experiments in these rich systems could cast further light on these enigmatic entities and provide an inspiring context for their properties in the cosmos.

## Methods
### Order Parameter
The mean-field order parameter of a spin-$F$ BEC $\Psi(\mathbf{r}, t) = \sqrt{n(\mathbf{r},t)}\zeta(\mathbf{r},t)$ is a mapping from a region in physical space into an order parameter space consisting of elements with $2F + 1$ complex-valued spinor components. Different subspaces of the full spinor in U($2F + 1$) can be identified, such as the ferromagnetic and polar phases of the spin-1 BEC. Each subspace, or magnetic phase, has its specific symmetries that give rise to the spectrum of topological defects supported by the phase. The classification of the topological defects can be carried out using homotopy theory[35]. More details of the order parameter and homotopy groups are provided below.

### Hamiltonian density
The mean-field Hamiltonian density for a spin-1 Bose–Einstein condensate in an external magnetic field oriented along the $z$ axis can be written as[26]

$$\mathcal{H} = \frac{\hbar^2}{2M}|\nabla\Psi|^2 + n\left(U_{\text{trap}} + p\langle F_z\rangle + q\langle F_z^2\rangle\right) + \frac{n^2}{2}\left(c_0 + c_2|\langle\mathbf{F}\rangle|^2\right) \quad (3)$$

where $M$ is the atomic mass, $n = |\Psi|^2$ is the atomic density, and $\langle\mathbf{F}\rangle = \zeta^\dagger\mathbf{F}\zeta$ the local average spin obtained from the vector $\mathbf{F} = (F_x, F_y, F_z)$ of the standard spin-1 matrices $F_\alpha$. The strength of the harmonic optical trapping potential $U_{\text{trap}} = M(\omega_r r^2 + \omega_z z^2)/2$ is determined by the radial and axial trapping frequencies $\omega_r$ and $\omega_z$, respectively. The pairwise interaction terms are $c_0 = [4\pi\hbar^2/(3M)](2a_2 + a_0)$ and $c_2 = [4\pi\hbar^2/(3M)](a_2 - a_0)$, where $a_\mathcal{F}$ is the s-wave scattering length in the total spin-$\mathcal{F}$ channel of two atoms. For $^{87}$Rb, these scattering lengths[36] are $a_0 = 101.8(2)a_B$ and $a_2 = 100.4(1)a_B$ in terms of the Bohr radius $a_B$. The linear Zeeman energy shift is $p = g\mu_B B_z$, where $g$ is the Landé g-factor and $\mu_B$ is the Bohr magneton. The quadratic Zeeman shift is given by $q = (g\mu_B B_z)^2/\Delta E > 0$ in terms of the ground-state hyperfine splitting $\Delta E \approx 6.8$ GHz.

### Magnetic phases
The two possible ground-state magnetic phases of a spin-1 BEC are characterised by the local spin expectation value. In the ferromagnetic phase, $|\langle\mathbf{F}\rangle| = 1$ and in the polar phase, $|\langle\mathbf{F}\rangle| = 0$. In the absence of an external magnetic field, the sign of the interaction term $c_2$ in the Hamiltonian determines the ground-state magnetic phase[37]. For $^{87}$Rb, $c_2 < 0$ and hence the ground-state phase is ferromagnetic, whereas in the presence of a field $|\mathbf{B}| \approx 1$ G the ground-state phase is polar with the director aligned with the magnetic field as a result of the quadratic Zeeman shift $nq\langle F_z^2\rangle$ (see Eq. (3)). The more general mixed phase consists of coexisting ferromagnetic and polar order. Continuity of the total order parameter guarantees that the topological analysis given

throughout this paper applies not only to the pure polar phase but also to any nonvanishing polar order in the mixed phase.

## Symmetry of the Hamiltonian density

With no external magnetic field, the Hamiltonian density in equation (3) is invariant under both global SO(3) spin rotations and shifts of the scalar phase, which together form the elements of the group $G = \mathrm{SO}(3)_{\mathbf{f}} \times \mathrm{U}(1)_{\varphi}$ with subscripts $\mathbf{f}$ and $\varphi$ referring to the spin and scalar-phase degrees of freedom. The group elements can be represented by $e^{i\varphi}U(\alpha, \beta, \gamma) = e^{i\varphi}e^{-i\alpha F_z}e^{-i\beta F_y}e^{-i\gamma F_z}$, where $\alpha, \beta,$ and $\gamma$ are Euler angles and $\varphi$ is a scalar phase.

## Order parameter space

The action of $G$ on the representative polar spinor $\tilde{\zeta}_{\mathrm{P}} = (0, 1, 0)^{\mathrm{T}}$ yields the arbitrary polar spinor $\zeta_{\mathrm{P}}$,

$$\zeta_{\mathrm{P}} = e^{i\varphi}U(\alpha, \beta, \gamma)\tilde{\zeta}_{\mathrm{P}} = \frac{e^{i\varphi}}{\sqrt{2}}\begin{pmatrix} -e^{-i\alpha}\sin\beta \\ \sqrt{2}\cos\beta \\ e^{i\alpha}\sin\beta \end{pmatrix} \quad (4)$$

as expressed in a basis quantized along $z$. The set of distinct $\zeta_{\mathrm{P}}$ constitutes the polar order parameter space $\mathcal{M}_{\mathrm{P}}$. Since $\gamma$ does not appear in equation (4), $\zeta_{\mathrm{P}}$ is invariant with respect to SO(2) rotations about the nematic axis specified by $\alpha$ and $\beta$. It is also invariant with respect to the discrete $\mathbb{Z}_2$ spin-phase coupling symmetry produced by the simultaneous transformations $\varphi \to \varphi + \pi$ and $\beta \to \beta + \pi$. These invariant transformations form a subgroup $H = \mathrm{SO}(2)_{\mathbf{f}} \rtimes (\mathbb{Z}_2)_{\mathbf{f},\varphi}$ of $G$, signaling that the symmetry of the polar phase is broken from $G$. As a result, the polar order parameter space is[38]

$$\mathcal{M}_{\mathrm{P}} = \frac{G}{H} = \frac{\mathrm{SO}(3)_{\mathbf{f}} \times \mathrm{U}(1)_{\varphi}}{\mathrm{SO}(2)_{\mathbf{f}} \rtimes (\mathbb{Z}_2)_{\mathbf{f},\varphi}} \cong \frac{S_{\mathbf{f}}^2 \times \mathrm{U}(1)_{\varphi}}{(\mathbb{Z}_2)_{\mathbf{f},\varphi}} \quad (5)$$

## Nematic director

Equation (5) suggests that any $\zeta_{\mathrm{P}}$ can be represented in terms of a point on the $S^2$ unit sphere and a U(1) phase on the unit circle, with opposite points on the sphere identified. Rewriting equation (4), we find

$$\zeta_{\mathrm{P}} = \frac{e^{i\varphi}}{\sqrt{2}}\begin{pmatrix} -d_x + id_y \\ \sqrt{2}d_z \\ d_x + id_y \end{pmatrix} \quad (6)$$

where the director $\hat{\mathbf{d}} = (\sin\beta\cos\alpha, \sin\beta\sin\alpha, \cos\beta)$ picks the point on the unit sphere. Thus the spinor representation $\zeta_{\mathrm{P}}(\mathbf{r}, t) = e^{i\varphi(\mathbf{r},t)}\hat{\mathbf{d}}(\mathbf{r}, t)$ maps vectors in the physical space to the order parameter space $\mathcal{M}_{\mathrm{P}}$ as characterised by $\varphi$ and $\hat{\mathbf{d}}$. The discrete $\mathbb{Z}_2$ symmetry corresponds to the relation $e^{i\varphi}\hat{\mathbf{d}} = e^{i(\varphi+\pi)}(-\hat{\mathbf{d}})$.

The projections $\hat{\mathbf{p}} = \hat{\mathbf{x}}$ and $\hat{\mathbf{p}} = \hat{\mathbf{y}}$ place the $x$ and $y$ components of the director, respectively, into the $m = 0$ component of the spinor with respect to the basis quantized along $\hat{\mathbf{p}}$. For example, the projection $\hat{\mathbf{p}} = \hat{\mathbf{x}}$ yields the relation

$$\zeta_{\mathrm{P}} = \frac{e^{i\varphi}}{\sqrt{2}}\begin{pmatrix} -d_y + id_z \\ \sqrt{2}d_x \\ d_y + id_z \end{pmatrix}_x \quad (7)$$

as can be seen by appropriately permuting the variables of Eq. (6). Thus the $m = 0$ spinor component densities with different projection axes yield the squares of the corresponding components of $\hat{\mathbf{d}}$ (see, for example, Fig. 2g–i).

A Cartesian spin-basis decomposition[26] extracts the nematic order $\varphi$ and $\hat{\mathbf{d}}$ from a general mixed-phase spinor $\zeta(\mathbf{r}, t)$. Following ref. 39, the spinor is written as $\zeta = \mathbf{u} + i\mathbf{v}$, in terms of a pair of real vectors $\mathbf{u}$ and $\mathbf{v}$ (with $|\mathbf{u}|^2 + |\mathbf{v}|^2 = 1$). The phase $\varphi$ is then calculated by applying the scalar-phase transformation $\zeta = e^{i\varphi}e^{-i\varphi}(\mathbf{u} + i\mathbf{v}) = e^{i\varphi}(\mathbf{a} + i\mathbf{b})$ such that $\mathbf{a} \cdot \mathbf{b} = 0$ and $|\mathbf{a}| \geq |\mathbf{b}|$, i.e., by finding the corresponding solution of $\tan 2\varphi = 2\mathbf{u} \cdot \mathbf{v}/(|\mathbf{u}|^2 - |\mathbf{v}|^2)$. The ferromagnetic order is defined by $\langle \mathbf{F} \rangle = 2(\mathbf{a} \times \mathbf{b})$. For $|\mathbf{a}| > |\mathbf{b}|$ the nematic order is defined by $\varphi$ and $\hat{\mathbf{d}} = \mathbf{a}/|\mathbf{a}|$, matching those of the pure polar spinor $\zeta_{\mathbf{P}}$ if $\mathbf{b} = \mathbf{0}$ (and therefore $|\langle \mathbf{F} \rangle| = 0$). For $|\mathbf{a}| = |\mathbf{b}| = \frac{1}{\sqrt{2}}$ the director is undefined and the phase is purely ferromagnetic, with $|\langle \mathbf{F} \rangle| = 1$.

## Topological defects

Topological defects within a medium are classified according to homotopy theory, which provides a way to determine which defects are topologically stable, i.e., cannot be continuously deformed into the uniform field configuration[35]. The first two homotopy groups $\pi_1$ and $\pi_2$ characterise line and point defects, respectively. In the case of the polar phase BEC, $\pi_2(\mathcal{M}_{\mathrm{P}}) = \mathbb{Z}$, and thus monopoles are topologically stable with an integer topological charge. Vortices, $\pi_1(\mathcal{M}_{\mathrm{P}}) = \mathbb{Z}$, are also topologically stable in the polar phase, with either integer or half-integer winding. The fact that the polar phase supports both $\pi_1$ and $\pi_2$ defects enables the monopole core topology to change locally to a vortex ring.

## Experimental methods

Our experimental apparatus and the monopole creation process have been described previously in ref. 25. In brief, the experiment involves a $^{87}$Rb condensate of approximately $2.5 \times 10^5$ atoms confined in a 1064 nm crossed-beam optical dipole trap with radial and axial frequencies of 120 Hz and 160 Hz, respectively. The condensate is exposed to the magnetic field $\mathbf{B}(\mathbf{r}, t)$ specified by equation (2), which is produced by three orthogonal pairs of Helmholtz coils that generate the bias field contribution $\mathbf{B}_{\mathrm{b}}$, and a pair of anti-Helmholtz coils that generates the spherical quadrupole field contribution $\mathbf{B}_{\mathrm{q}}$ with an adjustable amplitude $b_{\mathrm{q}}$.

The condensate is prepared in the polar $|F = 1, m = 0\rangle$ state with $\hat{\mathbf{d}} = \hat{\mathbf{z}}$ oriented along a uniform magnetic field of strength 1.0 G. The quadrupole contribution to the magnetic field is then initialised to $b_{\mathrm{q}} = 4.3$ G/cm and the bias contribution reduced to $B_{\mathrm{b}} = 45$ mG along $+\hat{\mathbf{z}}$. Importantly, $\hat{\mathbf{d}}$ remains well aligned with the magnetic field throughout the condensate. These parameters place the zero of the magnetic field 52 μm above the centre of the condensate. The bias field contribution is then reduced by a constant rate $\dot{B}_{\mathrm{b}} = -0.25$ G/s for 180 ms, bringing the field zero to the centre of the condensate. The monopole is imprinted in the $\hat{\mathbf{d}}$ field since, in the spirit of the adiabatic theorem, the directors adiabatically follow the magnetic field lines during this creation ramp. Straightforward modifications to this protocol can be used for different orientations of $\mathbf{B}_{\mathrm{b}}$ immediately prior to the creation ramp, which has the effect of changing the direction along which the field zero enters the condensate[25].

Immediately after the monopole is created, the quadrupole field is quickly (in 30 μs) extinguished and $\mathbf{B}_{\mathrm{b}}$ is rapidly (in 40 μs) increased to 1.2 G along the chosen projection direction $\hat{\mathbf{p}}$, which in these experiments is either $+\hat{\mathbf{z}}$, $-\hat{\mathbf{x}}$, or $+\hat{\mathbf{y}}$. The condensate is then held in the constant bias field of 1.2 G for a time $T$, after which the optical trap is extinguished and the condensate expands for 23 ms. Several milliseconds after the release, the cloud is exposed to a brief magnetic-field gradient which causes the spinor components to separate along the $x$ axis. The components are imaged absorptively along the $z$ and $y$ axes.

All different experimental images shown in this paper are representative examples of hundreds of individual results obtained under similar circumstances.

## Numerical simulations

We simulate the BEC dynamics by numerically integrating the three-dimensional spin-1 Gross–Pitaevskii equations[40]. The simulation begins by finding the BEC density profile in the ferromagnetic ground state in zero magnetic field. At each point of the computational grid

the ferromagnetic spinor is then replaced by the polar spinor $(0, 1, 0)^T$ and perturbed by a noise term $0.1 \times (a + ib, c + id, e + if)^T$, where $a...f$ are random real numbers with a Gaussian distribution with a mean of zero and a standard deviation of one. The order parameter is subsequently normalised across the entire condensate.

The simulation follows the experimental protocol from the beginning of the creation ramp to the release from the optical trap. The free expansion is simulated with a ballistic approximation[41] and the resulting column density profiles are determined individually for each spinor component, omitting the Stern–Gerlach separation pulse. We take into account the three-body loss rate[42], $\alpha = \Gamma/\hbar\omega_r$ with $\Gamma = \hbar \times 2.9 \times 10^{30}\,\mathrm{cm}^6/\mathrm{s}$. We employ a cubic computational domain of $256^3$ points, with pre-expansion side length $L = 23\,\mu\mathrm{m}$ and post-expansion side lengths $L_r = 328, L_z = 511\,\mu\mathrm{m}$, and the results converge. A second numerical integration method based on discrete exterior calculus[43] is used for the long-time evolution.

### Healing lengths

The size of the monopole core is determined by the density and spin healing lengths, $\xi_n = 0.17\,\mu\mathrm{m}$ and $\xi_s = 2.5\,\mu\mathrm{m}$, respectively. Since $\xi_s > \xi_n$ it can be energetically favourable to fill singularities with superfluid of a different magnetic phase. The characteristic size of these filled regions is determined by $\xi_s$, which is approximately one third of the size of the condensate ($R_{\mathrm{TF}} = 7.6\,\mu\mathrm{m}$) and therefore readily observable.

### Half-quantum vortex dipole

The polar phase supports half-quantum vortices that can have filled ferromagnetic cores[23,44]. As a straightforward example, we consider the superposition of the $m = \pm 1$ spinor components

$$\zeta_{\mathrm{hqv}}(\phi) = \frac{1}{\sqrt{2}} \begin{pmatrix} e^{i\phi} \\ 0 \\ 1 \end{pmatrix} = \frac{e^{i\phi/2}}{\sqrt{2}} \begin{pmatrix} e^{i\phi/2} \\ 0 \\ e^{-i\phi/2} \end{pmatrix} \qquad (8)$$

where $\phi$ is the azimuthal coordinate taken about the vortex line in the $m = +1$ spinor component. This spinor describes, for example, one of the two filled-core vortices observed experimentally in the $xy$ plane of the condensate (Fig. 4b, e). We can rewrite $\zeta_{\mathrm{hqv}}$ in terms of the director and quantum phase,

$$\hat{\mathbf{d}}_{\mathrm{hqv}}(\phi) = \left(\cos\frac{\phi}{2}, -\sin\frac{\phi}{2}, 0\right) \qquad \text{and} \qquad \varphi_{\mathrm{hqv}}(\phi) = \frac{\phi}{2} \qquad (9)$$

This is a half-quantum vortex, where $\varphi$ changes by $\pi$ and $\hat{\mathbf{d}}$ experiences a $\pi$ disclination in the range $\varphi \in [0, \pi)$. A similar analysis with the $m = \pm 1$ spinor components interchanged and $\varphi \to -\varphi$ yields a half-quantum vortex with the opposite circulation and core magnetisation. Thus the filled-core vortex pair visible in Fig. 4 constitutes a half-quantum vortex dipole, as is expected for the intersection of a half-quantum vortex ring with the $xy$ plane.

## Data availability

The experimental data and simulation results generated in this study have been deposited in the Zenodo database under accession code https://doi.org/10.5281/zenodo.8027653, ref. 45.

## Code availability

The simulation code can be provided by the authors upon request.

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

## Acknowledgements
We acknowledge financial support from the National Science Foundation through Grant Nos. PHY–1806318 and PHY–2207631 (D.S.H.), and from the Academy of Finland through its Centre of Excellence in Quantum Technology Grant No. 336810 (M.M.).

## Author contributions
A.B., T.O., and D.S.H. developed and conducted the experiments and analysed the data. Numerical simulations were carried out by R.Z.-Z. and M.K. under the supervision of M.M. All authors discussed the results and commented on the manuscript.

## Competing interests
M.M. declares that he is a Co-Founder and Shareholder of IQM Quantum Computers and a Research Professor at VTT Technical Research Centre of Finland Ltd. The other authors declare no competing interests.
