## [Peer Review File · Nature Communications]

Observation of an Alice Ring in a Bose-Einstein CondensateREVIEWER COMMENTS

Reviewer #1 (Remarks to the Author):

The authors experimentally constructed an Alice ring in an ultracold atomic gas, more precisely in an $F=1$ spinor Bose-Einstein condensate (BEC) in the polar phase. They first created a monopole in this system and observed a decay into an Alice ring. They found a fair agreement between experiments and simulations. Alice strings are exotic objects originally proposed in high energy physics about forty years ago. Realization in experiments is very exciting and novel. The manuscript is carefully written.

However, there are several points that should be clarified before I reconsider this paper for publication in Nature communications. They should address the following comments and questions.

(1) The theoretical prediction of this work in the $F=1$ spinor BECs was first presented in Ref.[14] and so this reference should be more appreciated.

(2) The authors cite Refs.[10,11] for a core decay of a monopole. These references discuss a monopole core decay into a vortex ring in an $SU(2)$ gauge theory with a 5 representation (real five scalar fields, or traceless symmetric 3×3 tensor). Such a decay happens in the uniaxial phase -- the case in which Schwartz originally considered Alice strings ---. However, these are not the first references. The order parameter manifold is precisely the same with the one of uniaxial nematic liquids.

Actually, such a decay was already pointed out in nematic liquid crystals long time ago: H. Mori and H. Nakanishi, J. Phys. Soc. Jpn. 57, 1281-1286 (1988), <https://doi.org/10.1143/JPSJ.57.1281>; H. Nakanishi, K. Hayashi and H. Mori, Commun.Math. Phys. 117, 203–213 (1988). <https://doi.org/10.1007/BF01223590> (and probably earlier by Mineev in 1980). Refs. [10,11] just did the same in a gauge theory. The authors should acknowledge this fact.

(3) As one of the peculiar properties of Alice strings, the authors emphasized the fact that an Alice string converts a monopole into an anti-monopole as it travels around the string. In fact, they say in the end of abstract “Importantly, our work paves the way for the experimental discovery of the monopole antimonopole conversion associated with Alice strings” and they concluded the paper by saying “Long-lived Alice rings may enable the demonstration of their charge conjugation property on monopoles.”

However, this is questionable because in their setup, monopoles must decay into Alice rings. Thus, does this scenario imply that one Alice ring must travel inside the Alice ring of the same size? How is it possible?

(4) The authors also suggested in discussion analogues of the Alice ring in the uniaxial nematic phase of an $F=2$ spinor BEC. This is related to the above comment (2). It is, however, known that the ground state of nematic $F=2$ BECs has continuous degeneracy, which can be lifted slightly by quantum effects (Son et. al., Phys.Rev.Lett. 98 (2007) 160408; S. Uchino et. al. Phys.Rev.Lett. 105 (2010) 230406). The (approximate) order parameter space is thus larger. I wonder if this proposal is possible.

Reviewer #2 (Remarks to the Author):

The authors of the manuscript have experimentally and numerically studied the subject of Alice string/(polar) Monopole in cold condensates. Alice string as a theoretical concept has fascinated many theorists for many years and it has exotic properties including monopole-anti-monopole conversion.

I appreciate both the experimental and numerical efforts in this manuscript to push forward along this direction. Although the basic concept is relatively simple and transparent as illustrated in Fig.1, I have spent quite a bit time to understand details in experiments (as well as numerical calculations) presented in the manuscript. Apparently, while certain groups of data are in good agreement with schematics or conventional paradigms of topological defects and look quite promising, there are also groups of data that are confusing at least, not fully consistent with the standard paradigm of defect theory. They may also be related to certain misleading statements about polar condensates.

Below I will summarize two main concerns about the results/interpretations.

1) As illustrated in the manuscript, the difference between monopoles and Alice strings are local, not global. According to the manuscript, Fig.2c, 2d (and Fig.2e, 2f) are presented for an Alice string, and Fig.2a, b are for a monopole. It has a very nice presentation about $m=0$ state. However, if a reader takes the data presented literally, the reader will find $+1$, -1 states in Fig. 2a, b are globally different Fig. 2c, Fig 2.e and the difference proliferates over a whole condensate rather than over the core region of Alice string which I believe it is much smaller than the condensate. Namely, the upper half ($z>0$) and lower half ($z<0$) now become partially polarized along the z -direction according the colour scheme. That is not a characteristic of a polar phase. Neither does it look like a standard far field of a topological defect in a polar state. Readers will have tough time to make a connection with the standard paradigm. In fact, what Fig 2c, Fig.2e seem to suggest is a very different scenario when compared to the standard polar state defect picture. It appears to indicate more complicated dynamics than the simple creation dynamics of an Alice string near the centre that the authors had intended to focus on.

2) Perhaps somewhat related issues are discussions on the effect of ferromagnetic interactions on dynamics of a polar state in the manuscript. Typically, interactions play a very important role when ground states and defects are concerned. They also are crucial to quantum dynamics. There had been many previous efforts specifically looking into spin dynamics for different initial states.

Here, the authors initialized the condensate in a polar state although the ground state of Rb atoms is believed to be ferromagnetic. In other words, the polar manifold created is an excited manifold which in principle can be subject to further quantum spin dynamics. Naively, one shall expect an excited manifold is somehow unstable. For instance, phase separation can further lower interaction energies while the total magnetization is held fixed. A simple global-conservation-law usually is insufficient to prevent a polar state from evolving into phase separated ferromagnetic domains. [the statement of "...in the presence of $B=1\text{G}$ and conserved magnetisation, the ground-state phase is polar. " appears to be quite misleading.] In the manuscript, I am not able to find discussions on quantum dynamics of the highly excited polar manifold without an imprinted monopole.

Generally, it is very difficult for an excited many-body state state not to evolve into other states outside the initial manifold.

The authors can explore the following questions:

Are ferromagnetic interactions trivial so that do not play a critical role in the defect dynamics in an excited polar manifold studied here? (This is apart from the nucleation of Alice string that the authors had intended to focus on.) If indeed not, that is a very surprising result from a general dynamic point of view and the authors shall definitely elaborate on the mechanism.

If yes, what role do they actually play in defect dynamics? And how will the Alice string dynamics here in an excited polar manifold (with ferromagnetic interactions) be different from the one in a ground state manifold (with antiferromagnetic interactions, say for Na)?

Reviewer #3 (Remarks to the Author):

Below, I provide an assessment of NCOMMS-22-33672-T using the guideline from <https://www.nature.com/nature-portfolio/editorial-policies/peer-review#general-information>

Key Results: time evolution (observed experimentally and calculated numerically) of a magnetic monopole defect. The manuscript continues the line of research presented in Ref. 12 [Science 348, 544–

547 (2015)] by some of the same authors. In that work a monopole was created and studied in a spin-1 Bose-Einstein condensate.

Validity: It seems that experimental and numerical results are valid. My judgement is based on the fact that these results rely on well-established methods, and appear reasonable.

Originality: The results appear original. I am not aware of any work that presented these data.

Significance: In my subjective opinion, the manuscript presents only an incremental advance in the field and will be of interest only to some condensed matter physicists. Numerical simulations follow previous studies about Alice strings in spin-1 Bose Einstein condensates see papers [19,14] and review [17]. Experimental results only slightly (my subjective opinion) extend previous studies of topological defects.

Methodology: The manuscript relies on previous methods discussed elsewhere. These methods seem suitable for the problem at hand.

Data: The manuscript could improve presentation to ensure reproducibility of the results. For example,

1) it is not clear if the experimental data are taken after a single experimental run or they are some average over many runs.

2) temperature and its role are not discussed.

3) the methods contain the following statement: “perturbed by a noise term ... with a Gaussian distribution”. I guess that the mean of the distribution is zero and the standard deviation is 1. In any case, the manuscript should contain this information.

4) more information about the extended data would do the manuscript good. What is meant by “Monopole created from the $+\hat{z}$, $-\hat{z}$ and $+\hat{x}$ directions”. Is it related to the direction of the initial magnetic field B_b ?

Appropriate use of statistics and treatment of uncertainties: The manuscript does not discuss accuracy of the experimental and numerical data. For example, it is not clear if there are any error-bars in Extended Data Fig. 2. Also, it is not clear if the numerical results are converged for the chosen grid.

Conclusions: The main finding of the manuscript is presented in Fig. 2. The density of the condensate develops a ‘column’ (see Fig. 2 c), which is interpreted as an Alice string. This interpretation is consistent with numerical simulations. However, it is unclear to me if this interpretation is unique provided data

scarcity. For example, the simplest interpretation would be that the condensate simply fills the hole (vortex core) during the expansion. In my opinion, the manuscript does not provide a strong experimental evidence that the observed dynamics corresponds to an Alice string. This defect is not on my research agenda, therefore, I cannot suggest a smoking gun measurement that would tell an Alice string from other physics. However, I strongly believe that the manuscript should provide more information to validate their interpretation.

Suggested improvements (besides the improvements suggested above):

The role and significance of some parameters in the manuscript are not discussed. For example,

a) what is the role of the three-body term in the numerical simulations? What will happen if it is turned off or its value is different.

b) why is $B_b=1.2$ after the monopole is created? This value of the magnetic field leads to a polar ground state. Is it important? What would happen for other values of B_b ?

c) what do we learn from Figs. 2g-i? Why are they important?

d) the paper states that “the dissipative decay” is important in the introduction. However, the role of dissipation is not discussed in the main text.

e) in general the role of beyond-mean-field physics (finite temperature, three-body decay, dissipation) is not discussed well.

Clarity and context: I believe that in general the manuscript is well written. However, it might be hard to read for a non-specialist, as it strongly relies on the knowledge of previous literature.

Response to Reviewer #1.

We thank the Reviewer for their close reading of the manuscript.

The authors experimentally constructed an Alice ring in an ultracold atomic gas, more precisely in an $F=1$ spinor Bose-Einstein condensate (BEC) in the polar phase. They first created a monopole in this system and observed a decay into an Alice ring. They found a fair agreement between experiments and simulations. Alice strings are exotic objects originally proposed in high energy physics about forty years ago. Realization in experiments is very exciting and novel. The manuscript is carefully written. However, there are several points that should be clarified before I reconsider this paper for publication in Nature communications. They should address the following comments and questions.

We thank the reviewer for these positive comments, and are particularly gratified that they consider the realization of Alice rings “very exiting and novel.”

(1) The theoretical prediction of this work in the $F=1$ spinor BECs was first presented in Ref.[14] and so this reference should be more appreciated.

We have elevated the visibility of this reference (now [23]) by naming the authors within the main text (line 77). We also added a reference to some of Ruostekoski’s more recent work, which includes the possibility of an Alice ring in the biaxial nematic phase of a spin-2 Bose–Einstein condensate [24].

(2) The authors cite Refs.[10,11] for a core decay of a monopole. These references discuss a monopole core decay into a vortex ring in an $SU(2)$ gauge theory with a 5 representation (real five scalar fields, or traceless symmetric 3×3 tensor). Such a decay happens in the uniaxial phase – the case in which Schwartz originally considered Alice strings —. However, these are not the first references. The order parameter manifold is precisely the same with the one of uniaxial nematic liquids. Actually, such a decay was already pointed out in nematic liquid crystals long time ago: H. Mori and H. Nakanishi, J. Phys. Soc. Jpn. 57, 1281-1286 (1988), <https://doi.org/10.1143/JPSJ.57.1281>; H. Nakanishi, K. Hayashi and H. Mori, Commun.Math. Phys. 117, 203-213 (1988). <https://doi.org/10.1007/BF01223590> (and probably earlier by Mineev in 1980). Refs. [10,11] just did the same in a gauge theory. The authors should acknowledge this fact.

We now highlight these earlier results in the introduction as references [15-18].

(3) As one of the peculiar properties of Alice strings, the authors emphasized the fact that an Alice string converts a monopole into an anti-monopole as it travels around the string. In fact, they say in the end of abstract "Importantly, our work paves the way for the experimental discovery of the monopole antimonopole conversion associated with Alice strings" and they concluded the paper by saying "Long-lived Alice rings may enable the demonstration of their charge conjugation property on monopoles." However, this is questionable because in their setup, monopoles must decay into Alice rings. Thus, does this scenario imply that one Alice ring must travel inside the Alice ring of the same size? How is it possible?

The reviewer raises an excellent point about our current setup, in which we cannot do these experiments. In the future, however, such experiments may become possible if monopoles can be arbitrarily stabilized against deformation by targeting with, e.g., a laser beam that locally manipulates the atomic interaction properties. We have nevertheless diminished the emphasis on this particular future experiment as it is well beyond the scope of the present work.

(4) The authors also suggested in discussion analogues of the Alice ring in the uniaxial nematic phase of an $F=2$ spinor BEC. This is related to the above comment (2). It is, however, known that the ground state of nematic $F=2$ BECs has continuous degeneracy, which can be lifted slightly by quantum effects (Son et. al., Phys.Rev.Lett. 98 (2007) 160408; S. Uchino et. al. Phys.Rev.Lett. 105 (2010) 230406). The (approximate) order parameter space is thus larger. I wonder if this proposal is possible.

The degeneracy is lifted decisively by the presence of an applied magnetic field. Although the resulting ground state is biaxial nematic, we have successfully created monopoles in the uniaxial nematic phase and are even now attempting to understand the core deformation process in this rich system. We believe as a result that this proposal is definitely possible.

Response to Reviewer #2.

The authors of the manuscript have experimentally and numerically studied the subject of Alice string/(polar) Monopole in cold condensates. Alice string as a

theoretical concept has fascinated many theorists for many years and it has exotic properties including monopole-anti-monopole conversion.

I appreciate both the experimental and numerical efforts in this manuscript to push forward along this direction. Although the basic concept is relatively simple and transparent as illustrated in Fig.1, I have spent quite a bit time to understand details in experiments (as well as numerical calculations) presented in the manuscript. Apparently, while certain groups of data are in good agreement with schematics or conventional paradigms of topological defects and look quite promising, there are also groups of data that are confusing at least, not fully consistent with the standard paradigm of defect theory. They may also be related to certain misleading statements about polar condensates.

We thank the Reviewer for these generally positive statements and their effort in following the details of our manuscript. We are especially grateful for pointing out confusing passages so that we can amend them.

We will address the remainder of the Reviewer's comments after restating them in their entirety.

Below I will summarize two main concerns about the results/interpretations.

1) As illustrated in the manuscript, the difference between monopoles and Alice strings are local, not global. According to the manuscript, Fig.2c, 2d (and Fig.2e, 2f) are presented for an Alice string, and Fig.2a, b are for a monopole. It has a very nice presentation about $m=0$ state. However, if a reader takes the data presented literally, the reader will find $+1$, -1 states in Fig. 2a, b are globally different Fig. 2c, Fig 2.e and the difference proliferates over a whole condensate rather than over the core region of Alice string which I believe it is much smaller than the condensate. Namely, the upper half ($z > 0$) and lower half ($z < 0$) now become partially polarized along the z -direction according the colour scheme. That is not a characteristic of a polar phase. Neither does it look like a standard far field of a topological defect in a polar state. Readers will have tough time to make a connection with the standard paradigm. In fact, what Fig 2c, Fig.2e seem to suggest is a very different scenario when compared to the standard polar state defect picture. It appears to indicate more complicated dynamics than the simple creation dynamics of an Alice string near the centre that the authors had intended to focus on.

2) Perhaps somewhat related issues are discussions on the effect of ferromagnetic interactions on dynamics of a polar state in the manuscript. Typically, interactions play a very important role when ground states and defects are concerned. They also are crucial to quantum dynamics. There had been many previous efforts specifically looking into spin dynamics for different initial states.

Here, the authors initialized the condensate in a polar state although the ground state of Rb atoms is believed to be ferromagnetic. In other words, the polar manifold created is an excited manifold which in principle can be subject to further quantum spin dynamics. Naively, one shall expect an excited manifold is somehow unstable. For instance, phase separation can further lower interaction energies while the total magnetization is held fixed. A simple global-conservation-law usually is insufficient to prevent a polar state from evolving into phase separated ferromagnetic domains. [the statement of "... in the presence of B=1G and conserved magnetisation, the ground-state phase is polar. " appears to be quite misleading.] In the manuscript, I am not able to find discussions on quantum dynamics of the highly excited polar manifold without an imprinted monopole. Generally, it is very difficult for an excited many-body state state not to evolve into other states outside the initial manifold.

The authors can explore the following questions:

Are ferromagnetic interactions trivial so that do not play a critical role in the defect dynamics in an excited polar manifold studied here? (This is apart from the nucleation of Alice string that the authors had intended to focus on.) If indeed not, that is a very surprising result from a general dynamic point of view and the authors shall definitely elaborate on the mechanism.

If yes, what role do they actually play in defect dynamics? And how will the Alice string dynamics here in an excited polar manifold (with ferromagnetic interactions) be different from the one in a ground state manifold (with antiferromagnetic interactions, say for Na)?

These are excellent points. We shall address the second and then the first.

Point two: It is first worth noting that at magnetic fields of approximately 1 G the ground-state phase is polar, and not ferromagnetic, even though the interatomic interactions are ferromagnetic. This is because the quadratic Zeeman term proportional to $q\langle F_z^2 \rangle$ in the Hamiltonian (Eq. 3) dominates the ferromagnetic term proportional to c_2 ; specifically, $q > 0$ and thus at sufficiently large fields the polar phase $(0, 1, 0)^T$ is preferred. We mentioned

the conservation of magnetization in this context simply because if magnetization is not conserved then in an applied magnetic field the Zeeman term dominates and the true ground state is ferromagnetic, described by $(1, 0, 0)^T$. In our scenario magnetization is conserved, and condensates prepared in $(0, 1, 0)^T$ at 1 G persist in that state for the full lifetime of ~ 1 s. We have also found that the presence of a magnetic field gradient inhibits the formation of ferromagnetic domains even at low fields [see Ollikainen et al., PRX **7**, 021023 (2017) and the Supplement to Ray et al., Science **348**, 544 (2015)], which is what enables us to create monopoles in the polar phase using the magnetic imprinting method. We regret that our statement concluding “... the ground-state phase is polar” was perceived as misleading. We have modified the manuscript to make clear the essential point, i.e., that the ground-state phase is in fact polar at these fields (lines 101–102 and 247–253).

Thus we would not say that the ferromagnetic interactions do not matter, but rather that the effect of the magnetic field matters more. In particular, the manifold may perhaps not be as highly excited as one might at first think.

With respect to the differences with sodium condensates, for which $c_2 > 0$, the effects are subtle. To a first approximation, at these fields, sodium condensates would prefer to be in equal superpositions of $m = \pm 1$ rather than in $m = 0$; this is the distinction between easy-axis polar (Rb) and easy-plane polar (Na) sub-phases. These are roughly equivalent in our experimental context, although the sodium experiments would not have to apply a bias magnetic field in order to retain a polar or antiferromagnetic ground-state phase. To a second approximation, the interatomic interactions play a further role in that they discourage overlap of some of the spinor components. For example, in sodium the $m = \pm 1$ spinor components prefer not to overlap with the $m = 0$ spinor component, whereas for rubidium the $m = \pm 1$ spinor components prefer not to overlap with each other. In the rubidium case this can enhance the presence of partially-polarized mixed-phase regions, to which we now turn in addressing the Reviewer’s first point.

Point one: The Reviewer is correct to note that the interior regions of the evolved monopole in Fig. 2 are partially polarized, which does not conform to the standard picture of a topological defect in the polar phase. We should have written more about this point. To summarize here, in an ideal world the condensate would remain in the pure polar phase, but for technical reasons we cannot shut off the magnetic field gradient instantaneously, and thus there is a brief period of time during which the $m = \pm 1$ spinor components experience differential magnetic forces along the z -axis that causes them to drift apart afterwards. The simulation includes these details and yields the good agreement between experiment and simulation shown in Fig. 2. By simulating instead a “perfect” turn-off of the magnetic gradient field, we can see that the $m = \pm 1$ spinor components do not separate in this fashion, confirming

the origin of the effect.

The imperfect coil turn-off moves the condensate locally out of the pure polar phase into a partially polarized mixed phase that has both ferromagnetic and polar order. Critically, the polar part of the mixed phase is also described by a director $\hat{\mathbf{d}}$ and a quantum phase φ that agrees with those defined in the pure polar phase. We explained the procedure by which this decomposition is achieved numerically in the Methods, but we did not do so in such a way to facilitate the understanding of the experimental observations shown in Fig. 2. We have now moved this explanation to a more prominent location in the Methods, and in the main text (lines 141–149) we now refer explicitly to the partially-polarized region as being in a mixed phase that retains polar order. We also discuss the matter in the Methods (lines 253–256 and 291–299).

We emphasize in the revised manuscript that the emergence of the partially polarized regions means that the polar order is camouflaged rather than destroyed (lines 141–149). In fact, as long as the mixed phase does not veer completely into a the fully polarized ferromagnetic phase we would expect the *topology* of the polar part to remain essentially the same as it would have been were the system to have remained entirely in the polar phase (lines 284–288), which is of course one of its great conceptual strengths. The proof of this inherited topology is that the appearance of the ferromagnetic part accompanies a continuous transformation of the polar part in terms of our decomposition, and hence it does not affect the topological considerations. (The exact spinor component composition will of course evolve differently.) The presence of ferromagnetic superfluid also makes it easier to fill the Alice ring in the polar superfluid. We think that the revised manuscript addresses these concerns more satisfactorily, and again note our gratitude for the Reviewer’s comments that have prompted us to make these changes and improvements.

Response to Reviewer #3.

Below, I provide an assessment of NCOMMS-22-33672-T using the guideline from <https://www.nature.com/nature-portfolio/editorial-policies/peer-review#general-information>

Key Results: time evolution (observed experimentally and calculated numerically) of a magnetic monopole defect. The manuscript continues the line of research presented in Ref. 12 [Science 348, 544-547 (2015)] by some of the same authors. In that work a monopole was created and studied in a spin-1 Bose-Einstein condensate.

Validity: It seems that experimental and numerical results are valid. My judgement

is based on the fact that these results rely on well-established methods, and appear reasonable.

Originality: The results appear original. I am not aware of any work that presented these data.

We thank the Reviewer for this initial analysis of our manuscript, which reflects the care we took in designing and undertaking these experiments.

Significance: In my subjective opinion, the manuscript presents only an incremental advance in the field and will be of interest only to some condensed matter physicists. Numerical simulations follow previous studies about Alice strings in spin-1 Bose Einstein condensates see papers [19,14] and review [17]. Experimental results only slightly (my subjective opinion) extend previous studies of topological defects.

We respectfully disagree. The first observation of an Alice ring (taking into account all known physical systems) is more than an incremental advance. Thus we think that its experimental realization will be of interest to a much wider variety of physicists, including those who work in grand unified field theories.

Methodology: The manuscript relies on previous methods discussed elsewhere. These methods seem suitable for the problem at hand.

Data: The manuscript could improve presentation to ensure reproducibility of the results. For example, 1) it is not clear if the experimental data are taken after a single experimental run or they are some average over many runs.

We thank the Reviewer for highlighting this omission. In fact, the data included in the manuscript are single shots that are typical of hundreds of similar measurements taken over a period of many months. We have included a note to this effect in the revised manuscript (lines 337–338) and in the caption to Fig. 2.

2) temperature and its role are not discussed.

We have chosen to include only a zero-temperature, mean-field analysis, as is typical for these sorts of experiments. We now mention this explicitly in the results section of the manuscript (line 57–58).

3) the methods contain the following statement: "perturbed by a noise term ... with a Gaussian distribution". I guess that the mean of the distribution is zero and the standard deviation is 1. In any case, the manuscript should contain this information.

The Reviewer surmises the situation correctly, and we have included this information in the revised manuscript (line 342–345).

4) more information about the extended data would do the manuscript good. What is meant by "Monopole created from the +z, -z and +x directions". Is it related to the direction of the initial magnetic field B_b ?

We thank the Reviewer for this comment, and have updated our description (lines 135–137) and the caption to Supplementary Figure 1 to make these concepts more clear.

Appropriate use of statistics and treatment of uncertainties: The manuscript does not discuss accuracy of the experimental and numerical data. For example, it is not clear if there are any error-bars in Extended Data Fig. 2.

We have re-analyzed our vortex data shown in Extended Data Fig. 2 (now Supplementary Fig. 3) using a new fitting function, and can now report the fit error bars associated with the data points. The accuracy of our other experimental and numerical data are treated in accordance with the norms of the field.

Also, it is not clear if the numerical results are converged for the chosen grid.

We have added a statement confirming that the numerical results do converge for the chosen grid.

Conclusions: The main finding of the manuscript is presented in Fig. 2. The density of the condensate develops a 'column' (see Fig. 2 c), which is interpreted as an Alice string. This interpretation is consistent with numerical simulations. However, it is unclear to me if this interpretation is unique provided data scarcity. For example, the simplest interpretation would be that the condensate simply fills the hole (vortex core) during the expansion. In my opinion, the manuscript does not provide a strong experimental evidence that the observed dynamics corresponds

to an Alice string. This defect is not on my research agenda, therefore, I cannot suggest a smoking gun measurement that would tell an Alice string from other physics. However, I strongly believe that the manuscript should provide more information to validate their interpretation.

We would hope that some of this criticism would abate given the existence of our extensive data set showing the primary features exhibited in Fig. 2. Although consistency with numerical simulations is one of the primary strengths of the manuscript, we have explicitly attempted to appeal as much as we can to the experimental observations within the framework of the accepted analytic theory rather than relying entirely on the simulations for the analysis.

We describe the behavior of the system within the general contours of the mean-field theory, homotopy theory, and topology, using the experimental and numerical techniques that we and others have developed over the past few decades. We stress that our Science paper [Ray et al., *Science* **348**, 544 (2015)] has extensively confirmed that the spinor structure of the monopole is as we claim. Taking this into account, and following the reasoning in the present text, the emergence of the Alice ring is a necessity.

We agree that the onus is definitely upon us to identify the smoking guns in a manner that is accessible to the widest audience. To restate them here: (1) the $m = 0$ column formation along the z -axis (lines 129–138); (2) the absence of other significant topological changes to the spinor component density distributions, including the absence of an $m = 0$ column along other projection directions (lines 150–160); and (3) the appearance of a half-quantum vortex dipole during the evolution of an initially displaced monopole (lines 186–202). The case is quite strong based only on these three observations and the analytic theory. The appearance of the Alice ring (or some related core deformation) is consistent with our observations and the theory. When we add in the numerical simulations we are confident the case becomes airtight.

With respect to filling in the vortex core during expansion, we note that in self-similar expansion vortex cores do not fill in as suggested. [See also our response to the Reviewer’s query about Figs. 2 g–i, below, and the analysis of Ray et al., *Science* **348**, 544 (2015).]

We thank the Reviewer for their comments because it they have helped us to see where our arguments have not been made clearly enough, and where we do need to include more information to assist with the interpretation. The text has been revised accordingly throughout the document.

Suggested improvements (besides the improvements suggested above): The role and significance of some parameters in the manuscript are not discussed. For example, a) what is the role of the three-body term in the numerical simulations? What will happen if it is turned off or its value is different.

The three-body term is included so that we match our experimental conditions as best as we can. Without its inclusion or for different values the experimental and simulated component density profiles would otherwise not match as well, but the essential topological physics would be unchanged.

b) why is $B_b=1.2$ after the monopole is created? This value of the magnetic field leads to a polar ground state. Is it important? What would happen for other values of B_b ?

We have chosen a bias magnetic field large enough that the ground state phase of the condensate is polar (lines 141–148, and also our response to Reviewer #1, above). Smaller values of \mathbf{B}_b will mean that we are no longer working with a condensate in a polar ground state. At that point we would be facing an abundance of complications possibly accompanied by the onset of a phase transition to the ferromagnetic phase, as suggested by Reviewer #2, above. We have endeavoured to make this point more clear in the revised text.

c) what do we learn from Figs. 2g-i? Why are they important?

We thank the Reviewer for bringing this point to our attention. The monopole can be decomposed into different bases, and in each basis there are two $m = 0$ lobes (with respect to the projection axis $\hat{\mathbf{p}}$) and countercirculating vortical regions with $m = \pm 1$ [lines 117–120; and see, for example, Ray et al., *Science* **348**, 544 (2015)]. For the projections shown in Figs. 2g–i, we do not see the column of $m = 0$ atoms. (Note, too, that if it were just a matter of the vortical regions being “filled in” during expansion then we would expect to see columns in the $m = 0$ spinor component independent of the projection direction.) We have made this point more clearly in the revised manuscript, as it is important to understanding why the absence of a column along the other projection axes implies something special about the column along the z axis (lines 120–121). We now explicitly introduce the idea that projections along the x or y directions put the corresponding component of the director vector into the $m = 0$ spinor component in that basis (lines 284–290). As a result, columns of $m = 0$ atoms appearing along other axes after the time-evolution would not be consistent with the creation of an Alice ring.

We have modified the discussion of this figure in order to make this explanation more intelligible.

d) the paper states that “the dissipative decay” is important in the introduction. However, the role of dissipation is not discussed in the main text.

This is a passing comment that echoes a point made by Ruostekoski and Anglin [Ruostekoski and Anglin, PRL **91**, 190402 (2003)]. We did not intend it to receive any great emphasis, although we believe that it is worthy of attention. Without additional theoretical tools, which we felt to lie beyond the scope of the manuscript, there is little we can add beyond pointing the reader to the relevant reference.

e) in general the role of beyond-mean-field physics (finite temperature, three-body decay, dissipation) is not discussed well.

We appreciate the desire to explore the system beyond the mean-field theory. However, we feel that beyond-mean-field effects are beyond the scope of the manuscript, and are not particularly relevant to the specific topological observations. We note that we do take the three-body loss term into account, which leads to dissipation, and the Gross–Pitaevskii equation also provides dynamics of the quasiparticle excitations.

Clarity and context: I believe that in general the manuscript is well written. However, it might be hard to read for a non-specialist, as it strongly relies on the knowledge of previous literature.

We thank the Reviewer for this comment, and have revised the manuscript with an aim to make it as accessible as possible to non-specialists.

WE THANK all three Reviewers again for their close attention to our manuscript, and for their comments and suggestions.

List of Changes

1. There are numerous stylistic and grammatical changes throughout the manuscript.

MAIN TEXT

2. We rewrote the abstract (lines 2–14) to remove references and conform to *Nature Communications* house style.
3. We extensively rewrote the introductory paragraphs (lines 16–51) to match the abstract and include the omitted references. We also included new references pertaining to monopole core deformation in liquid crystals.
4. We added a note about staying within the standard mean-field approximation (lines 57–58).
5. We added a note about the possibility of the condensate existing in a mixed-phase, with both polar and ferromagnetic characteristics (lines 63–65).
6. We named Ruostekoski and Anglin explicitly (reference [23], line 77).
7. We improved the language describing the phase change and director reversal that ensures continuity of the order parameter (lines 77–83).
8. We commented on the strength of the field that ensures that the polar phase is the ground state (lines 101–102).
9. We enhanced the discussion of the monopole spinor component densities to make clear that the lobe and vortex structures depend on the basis chosen by the projection direction (lines 109–121).
10. We brought discussion of the numerical results forward so that they appear before further descriptive analysis of the monopole from experimental images (lines 123–128).
11. We continued the description of the formation of the column, expanding on the lack of column in projection directions other than $\hat{\mathbf{z}}$, why the column along $\hat{\mathbf{z}}$ cannot be seen in the auxiliary projections, and comment on the interesting features that emerge when the magnetic field zero is brought into the condensate (lines 129–138).
12. In a new paragraph, we commented on the relative motion of the $m = \pm 1$ spinor components and the emergence of partially polarised regions. We note that the underlying topology is unchanged because the polar order is preserved even in these mixed-phase regions (lines 139–149).

13. We expanded our discussion of what it means for the monopole to be displaced from the center of the condensate, and provide a new figure with experimental images of displaced monopoles (Supplementary Figure 2), so as to underscore that the general topological features of the Alice ring match those of the undisplaced monopole (lines 150–160, and new Supplementary Figure 2).
14. We explained more fully the director orientation and phase jumps associated with the curve \mathcal{L} in Fig. 2 (lines 161–173).
15. We added to our remarks on how the direction from which the quadrupole field zero enters the condensate changes the sense of the circulation of the Alice ring, as shown in Supplementary Figure 1 (lines 182–185).
16. We further emphasized the importance of our findings in the Discussion (lines 209–214).

METHODS

17. We added a passage describing the mixed phase (lines 253–256).
18. We added a section explaining projection directions and how they can be interpreted in terms of the director $\hat{\mathbf{d}}$ (lines 284–290).
19. We moved the discussion of the Cartesian spin-basis decomposition to the section discussing the nematic director (lines 291–299).
20. We added a note about changing the direction from which the magnetic field zero enters the condensate (lines 326–329).
21. We added this note: “All different experimental images shown in this paper are representative examples of hundred of individual results obtained under similar circumstances.” (lines 337–338).
22. We added a comment on the Gaussian distribution of the initial noise (line 345).
23. We added a note confirming convergence (line 353).

FIGURES AND CAPTIONS

24. We added a note about the experimental images being representative of hundreds taken under similar circumstances to the caption of Fig. 2.
25. We added images to Supplementary Fig. 1 and extensively rewrote the caption to better show the effect of creating the monopole from different directions.

26. We created Supplementary Fig. 2 and its caption, which show and discuss some examples of monopoles created in locations displaced from the centre of the condensate.
27. We extensively re-analysed the data for (renumbered) Supplementary Fig. 3, which now includes error bars on each point and a slightly adjusted value for the estimated creation time of the off-centre Alice ring. We edited the caption to better explain our analysis.

All line numbers refer to the revised manuscript.

REVIEWER COMMENTS

Reviewer #1 (Remarks to the Author):

I'm mostly satisfied with the authors' reply and revised manuscript. However, they do not seem to properly reply to my comment on (4) about the degeneracy of the nematic phase in F=2 spinor BEC. More explanations and/or suitable revision is needed.

Reviewer #2 (Remarks to the Author):

I want to thank the authors for clarifying the nature of polar states in the specific magnetic fields they had worked with, the role of conserved magnetizations, and spin interactions. The revisions are very helpful, making the presentation more straight forward to follow.

In this new version, the authors have also explained clearly the origins of polarization in the far field of Alice string/monopoles due to a drift (in Fig 2). These additional discussions are definitely helpful for readers to understand what have been observed as well as their the limitations.

I agree that a polar state with finite polarization can be characterized by a complex vector, $u+iv$ where u and v are two real orthogonal vectors. However, if the far field order parameters are allowed to be complex vectors (in a five sphere) rather than a real vector (in a two sphere, the one usually assigned to a polar state without polarization), it is very challenging to argue monopoles or the far fields of Alice strings are topologically stable. That is because monopoles are well-defined only in a polar state manifold. When the far fields are further allowed to live in a much enlarged manifold such as a 5-sphere (up to $u(1)$ gauge and Z_2 group), topological monopoles discussed in the manuscript can in principle unwind itself into a trivial object. For instance, the 2nd homotopy group for 2-sphere of real directors is an integer group but such a group for a 5-sphere where complex vectors live is trivial, forbidding any topological states. [Z_2 group won't matter in this particular discussion of point defects.] This perhaps can be quite concerning from a concept point of view. Below are my suggestions.

1) I suggest adding the following discussion. If during the magnetic turn-off, drifting can't be practically avoided, can authors add the simulation results showing how an ideal world shall look like in this case, next to what had been seen in the experiment ? This will help to identify a direction of potential new efforts if other groups are interested in going beyond this or other related approaches to further improve.

2) It is promising to look for defects in other systems such as $F=2$ condensates which can have natural connections to $SU(2)$ gauge monopoles as Mentioned in Referee 1's report (point 2 and 4). The authors can include brief discussions along this line and comment on those two references mentioned by Referee 1 on $F=2$ condensates in point 4 (that indeed represent new progress in our understanding), to make the concluding remarks more exciting. The current discussion on future Opportunities appears to be a little bit thin.

I recommend the publication of the manuscript if point 1) and 2) are properly addressed.

Reviewer #3 (Remarks to the Author):

The revised manuscript addresses most of my comments satisfactory. It appears well written and significant to the field, hence I recommend it for publication.

Response to Reviewer #1.

I'm mostly satisfied with the authors' reply and revised manuscript. However, they do not seem to properly reply to my comment on (4) about the degeneracy of the nematic phase in $F=2$ spinor BEC. More explanations and/or suitable revision is needed.

For clarity, we reproduce Reviewer #1's original point (4) here:

(4) The authors also suggested in discussion analogues of the Alice ring in the uniaxial nematic phase of an $F=2$ spinor BEC. This is related to the above comment (2). It is, however, known that the ground state of nematic $F=2$ BECs has continuous degeneracy, which can be lifted slightly by quantum effects (Son et. al., Phys.Rev.Lett. 98 (2007) 160408; S. Uchino et. al. Phys.Rev.Lett. 105 (2010) 230406). The (approximate) order parameter space is thus larger. I wonder if this proposal is possible.

We misunderstood that the Reviewer was asking us to amplify these points in the manuscript itself. We now include the aforementioned citations, and we expand our concluding remarks to address the degeneracy of the $F = 2$ nematic phases. We also cite two references that precisely support our claim that, in addition to the effects of fluctuations, the two nematic phases are nondegenerate and therefore distinguishable in a nonzero magnetic field due to quadratic Zeeman shifts. We believe that these additions enrich the discussion, and we thank the Reviewer for insisting on clarification of this point.

Response to Reviewer #2.

I want to thank the authors for clarifying the nature of polar states in the specific magnetic fields they had worked with, the role of conserved magnetizations, and spin interactions. The revisions are very helpful, making the presentation more straight forward to follow.

We are pleased with the Reviewer's positive response.

In this new version, the authors have also explained clearly the origins of polarization in the far field of Alice string/monopoles due to a drift (in Fig 2). These

additional discussions are definitely helpful for readers to understand what have been observed as well as their the limitations.

I agree that a polar state with finite polarization can be characterized by a complex vector, $u+iv$ where u and v are two real orthogonal vectors. However, if the far field order parameters are allowed to be complex vectors (in a five sphere) rather than a real vector (in a two sphere, the one usually assigned to a polar state without polarization), it is very challenging to argue monopoles or the far fields of Alice strings are topologically stable. That is because monopoles are well-defined only in a polar state manifold. When the far fields are further allowed to live in a much enlarged manifold such as a 5-sphere (up to $U(1)$ gauge and Z_2 group), topological monopoles discussed in the manuscript can in principle unwind itself into a trivial object. For instance, the 2nd homotopy group for 2-sphere of real directors is an integer group but such a group for a 5-sphere where complex vectors live is trivial, forbidding any topological states. [Z_2 group won't matter in this particular discussion of point defects.] This perhaps can be quite concerning from a concept point of view. Below are my suggestions.

This is an excellent point. We agree that the monopoles can in principle unwind themselves in the expanded order parameter space, provided the mixed phase retains a partial polar character (excepting the region within the Alice ring). While we would greatly prefer to make a stronger claim, our argument is not that the monopole and Alice ring excitations are topologically stable *per se*, but that they do not unwind through this larger space during the time it takes to create and observe them. We now make the first point explicitly in the revised text at the top of page 8.

1) I suggest adding the following discussion. If during the magnetic turn-off, drifting can't be practically avoided, can authors add the simulation results showing how an ideal world shall look like in this case, next to what had been seen in the experiment ? This will help to identify a direction of potential new efforts if other groups are interested in going beyond this or other related approaches to further improve.

We thank the Reviewer for this suggestion and agree that the “ideal world” simulation helps to solidify our case, as well as to potentially bridge our work to future research. We have therefore provided an additional figure and caption, now Supplementary Figure 2 on page 20, that shows the “ideal world” simulation, in which the $m = \pm 1$ spinor components overlap well and thus largely preserve the polar phase. We make this point and refer to the new

figure in the text at the top of page 8.

2) It is promising to look for defects in other systems such as $F = 2$ condensates which can have natural connections to $SU(2)$ gauge monopoles as Mentioned in Referee 1's report (point 2 and 4). The authors can include brief discussions along this line and comment on those two references mentioned by Referee 1 on $F = 2$ condensates in point 4 (that indeed represent new progress in our understanding), to make the concluding remarks more exciting. The current discussion on future Opportunities appears to be a little bit thin.

We have enhanced and extended the concluding Discussion to consider the exciting prospect of Alice rings in an $F = 2$ BEC in both the uniaxial and biaxial nematic phases. As we note in our response to Reviewer #1, above, these phases are expected to be distinguishable due to the lifting of the degeneracy by both quantum fluctuations and the presence of a nonzero magnetic field. Our expanded discussion now includes the two references on the role of quantum fluctuations, as well as two additional references that consider the effects of the magnetic field.

I recommend the publication of the manuscript if point 1) and 2) are properly addressed.

We once again thank the Reviewer for their suggestions, and hope that they find points 1) and 2) are now addressed satisfactorily.

Response to Reviewer #3.

The revised manuscript addresses most of my comments satisfactory. It appears well written and significant to the field, hence I recommend it for publication.

We are pleased that the Reviewer recommends our manuscript for publication.

Finally, we thank all three Reviewers again for their close attention to our manuscript, and for their comments and suggestions.

List of Changes

MAIN TEXT

1. We added a reference to Supplementary Figure 2 (line 145).
2. We added a note about the potential instability of the monopole/ Alice ring excitation in the mixed-phase region (lines 146 - 147).
3. We clarified the language describing the continued existence of polar phase topology in the mixed-phase region (148 - 149).
4. We added a new paragraph to the Discussion in which we mention the degeneracy of the uniaxial and biaxial nematic $F = 2$ phases and discuss the prospect of the monopole and ring excitations in these phases (lines 222 - 234).
5. We included two new references in the Discussion pertaining to lifting the degeneracy of the uniaxial and biaxial nematic phases through quantum fluctuations (line 226).

FIGURES AND CAPTIONS

6. We added a new figure, now Supplementary Figure 2, showing a comparison between an “ideal world” simulation and the “experiment” simulation, making explicit the effect of the experimental magnetic field extraction on the BEC magnetization.

ACKNOWLEDGMENTS

7. We added our most recent NSF grant number (line 393).

All line numbers refer to the revised manuscript.

REVIEWERS' COMMENTS

Reviewer #1 (Remarks to the Author):

The authors satisfactory revised the manuscript.

Now I can recommend publication of this paper to Nature Communications.

Reviewer #2 (Remarks to the Author):

I want to thank the authors for improving the clarity of presentation. I recommend the publication in Nature communications.

I also further suggest to cite the following reference

Turner et al., Phys. Rev. Lett. 98, 190404 (2007)

(along with 32, 33 on the very similar subject.)

Response to Reviewer #1.

The authors satisfactory revised the manuscript. Now I can recommend publication of this paper to Nature Communications.

We thank the Reviewer for their recommendation.

Response to Reviewer #2.

I want to thank the authors for improving the clarity of presentation. I recommend the publication in Nature communications.

We are pleased to obtain the Reviewer's recommendation.

I also further suggest to cite the following reference

Turner et al., Phys. Rev. Lett. 98, 190404 (2007) (along with 32, 33 on the very similar subject.)

We now include this reference, which appears as reference [33] (sandwiched by the previous references [32] and [33]). The following references are re-numbered.

We thank all the Reviewers for the time and thought put into reviewing our work. We believe that the manuscript is improved as a result of their input.

LIST OF CHANGES

The following changes are highlighted in yellow in the PDF document Alice.Full.

Author list

The email address of corresponding author A.B. has been added.

Affiliations

M.M's affiliation has been edited to reflect current affiliations of the author.

Abstract

The specifier “ ^{87}Rb ” has been added to the description of the physical system for clarity to comply with the editor's request.

Main text

On page 7 “center” has been changed to “centre.”

Methods

On page 10 an extra “the” has been removed.

Data availability

The data availability statement has been modified to fit the editor’s guidelines, and a link to a publicly-accessible data repository has been added.

References

Reference [33] (*Turner et. al.*) has been added to comply with Reviewer’s request.

Acknowledgments

The funding acknowledgment has been modified to fit the editor’s guidelines.

Competing interests

The competing interests statement has been modified to more accurately state the author M.M.’s affiliations and positions within the affiliations.

Supplementary information

1. In Supplementary Figure 1 **c**, the bottom right panel has been replaced by the correct simulation image. Whilst preparing the data repository we noticed that the bottom-right simulation image was for $T = 6$ ms, as opposed to $T = 4$ ms as stated previously. Thus the correct simulation image, corresponding to $T = 4$ ms, has been inserted. Additionally, the caption has been changed to reflect the fact that the left column of **c** represents a BEC cloud at $T = 6$ ms, and not $T = 4$ ms as stated previously.
2. In the captions of Supplementary Figures 3 and 4, several instances of “center” have been changed to “centre.”